# Encapsulation of Imazalil in HKUST-1 with Versatile Antimicrobial Activity

**DOI:** 10.3390/nano12213879

**Published:** 2022-11-03

**Authors:** Hongqiang Dong, Yuke He, Chen Fan, Zhongqiang Zhu, Chunrong Zhang, Xinju Liu, Kun Qian, Tao Tang

**Affiliations:** 1State Key Laboratory for Managing Biotic and Chemical Threats to the Quality and Safety of Agro-Products, Key Laboratory for Pesticide Residue Detection of Ministry of Agriculture and Rural Affairs, Institute of Agro-Product Safety and Nutrition, Zhejiang Academy of Agricultural Sciences, Hangzhou 310021, China; 2College of Agriculture, Tarim University, Alaer 843300, China; 3College of Plant Protection, Southwest University, Chongqing 400716, China; 4Beijing Higher Institution Engineering Research Center of Food Additives and Ingredients, Beijing Technology and Business University, Beijing 100048, China

**Keywords:** imazalil, ionic liquid, HKUST-1, antimicrobial activity, safety

## Abstract

Based on high surface areas, adjustable porosity and microbicide activity, metal-organic frameworks (MOFs) HKUST-1 are widely used as drug release carriers for their slow degradation characteristics under slightly acidic conditions. In this work, porous HKUST-1 was reacted rapidly by cholinium salt (as the deprotonation agent and template) in an aqueous solution at room temperature. A novel antimicrobial system based on an imazalil encapsulated metal organic framework (imazalil IL-3@HKUST-1) was established. Imazalil IL-3@HKUST-1 could achieve synergism in inhibiting pathogenic fungi and bacteria. Moreover, six days after treatment, the slow and constant release of imazalil from imazalil IL@HKUST-1 exhibited better sustainability and microbicidal activity than imazalil. We believe that the method may provide a new strategy for related plant diseases caused by bacteria or fungi.

## 1. Introduction

As the traditional Chinese rhyme which says, “Hunger breeds discontentment”. Food is the first necessity of people, and food security is the basis of human survival and health. According to the latest news from the United Nations, the global population will reach an alarming 8.5 billion in 2030 [1]. High-quality agricultural products are necessary to feed billions of people and meet the growing demands for a better life. Although the use of pesticides or fungicides under field conditions plays an indispensable role in achieving high-quality and high-yield crops and agricultural products, there are also many shortcomings in the use of pesticides [2]. For example, the resistance of phytopathogenic microorganisms has become a problem, and the number of fungicides available is rapidly decreasing, resulting in reduced crop yield and quality. Some commercial pesticides may inadvertently cause harm to humans, other non-target organisms and the ecological environment [3,4]. In addition, as with some antifungal drugs, such as enilconazole (imazalil), tebuconazole, difenoconazole, etc., their efficacy is severely limited due to their poor penetration into plants. Therefore, it is urgent to develop a new strategy to overcome the antimicrobial resistance, environmental toxicity and poor efficacy of pesticides for sustainable plant disease management [5].

Metal–organic frameworks (MOFs), composed of metal nodes and organic linkers through coordination bonds, have attractive development prospects in modern materials research because of open unsaturated metal sites [6,7,8,9,10]. They have been widely used in gas storage, separation, catalysis and drug delivery [11,12,13,14]. In particular, HKUST-1 is attracting considerable attention due to its simple preparation procedure, low production costs, environmental compatibility, excellent chemical and thermal stability [15,16,17,18]. Furthermore, the Cu^2+^ released from HKUST-1 has antimicrobial properties against a variety of pathogens (such as *Pseudomonas syringae*, *Fusarium solani*, *Odontoglossum ringspot virus*, *Penicillium chrysogenum* and *Aspergillus niger*) [19,20]. More importantly, due to the large surface area, high porosity and tunable topologies, HKUST-1 is suitable to be used as a pesticide carrier [21].

Imazalil (4,6-dimethyl-*N*-phenylpyrimidin-2-amine), one of an anilinopyrimidine fungicides, has been widely used to control gray mold (*Botrytis cinerea*), rice blast (*Pyricularia oryzae*) and pear scab (*Venturia inaequalis*) [22]. However, imazalil is easily lost through volatilization (vapor pressure 2.2 mPa at 25 °C) [23]. In addition, imazalil is sensitive to sunlight, which results in its short half-life and low utilization efficiency in farming [24]. In recent years, the excellent characteristics of ionic liquid (IL) as a green alternative solvent to conventional organic solvents have raised considerable attention [25,26]; a realm where the amalgamation of biological activity and IL structure might bring extraordinary advantages in agrochemistry [27,28]. For example, IL-based herbicides could improve rain fastness; reduce the applied amount and environmental risk caused by leaching and runoff; and also enhance curative activity, penetration and persistence compared with non-IL form. Yang et al. dissolved abamectin as a functional additive with phosphorus-containing IL, and prepared core-shell microcapsules with gelatin cross-linking, which not only had slow-release characteristics, but also enhanced the solubility of abamectin and showed excellent insecticidal activity [29]. Therefore, it is a good choice to combine ILs with pesticides, especially with poorly soluble antifungal drugs to construct a novel pesticide.

Inspired by the reported literature, it is possible to design kinds of material-combined MOFs and IL-based pesticides. Herein, we developed a simple and fast synthesis strategy that resulted in HKUST-1 with cholinium salt as the deprotonation agent and template. Then, a novel antimicrobial system based on imazalil IL-encapsulated MOF (imazalil IL@HKUST-1) was established via the ship-in-bottle technique (Figure 1). The prepared nanocapsules were characterized by field emission scanning electron microscopy (SEM), X-ray photoelectron spectroscopy (XPS), N_2_ adsorption and thermogravimetric analysis (TGA); and then, we systematically investigated antibacterial activities against plant fungi and bacteria, genotoxicity and safety to plants.

## 2. Experiment Section

### 2.1. Materials

N,N-dimethylethanolamine, bromoethane, 1-bromooctane, 1-bromodecane, 1-bromododecane, 1,3,5-Benzenetricarboxylic acid and copper(II) chloride dihydrate were purchased from Aladdin Industrial Corporation (Shanghai, China). Imazalil was supplied by the Institute for the Control of Agrochemicals, Ministry of Agriculture (Beijing, China). Other chemicals were analytical grade and purchased from Sinopharm (Beijing, China). The products were prepared using ultra-pure water (Milli-Q system, Millipore Co., Billerica, MA, USA).

### 2.2. Synthesis Imazalil IL-3@HKUST-1

#### 2.2.1. Synthesis of N-ethyl-2-hydroxy-N,N-dimethylethanaminium Bromide

A solution of N,N-dimethylethanolamine (3.60 g) dissolved in ethanol (15 mL) was added to a solution of bromoethane (4.40 g) in ethanol (15 mL). After 24 h under reflux, the ethanol was removed by evaporation under vacuum. N-ethyl-2-hydroxy-N,N-dimethylethanaminium bromide: white solid, yield (97%), ^1^H NMR (300 MHz; D_2_O; Me_4_Si) δ ppm = 1.29 (t, 3H, -CH_3_), 3.04 (s, 6H, 2× -CH_3_), 3.38–3.41 (m, 4H, 2× -CH_2_-), 3.91 (br, 2H, -CH_2_-).

#### 2.2.2. Synthesis of Imazalil Ionic Liquid-3 (Imazalil IL-3)

Approximately 5.03 g of 1-bromododecane and 6.06 g of imazalil were dissolved in 40 mL of ethanol. After 48 h under reflux, the ethanol was removed by evaporation under vacuum. The products were washed with dichloromethane and ethyl acetate. Imazalil IL-3: brown liquid, yield (81%), ^1^H NMR (300 MHz; CDCl_3_; Me_4_Si).

#### 2.2.3. Synthesis of HKUST-1

Firstly, 1,3,5-benzenetricarboxylic acid (0.43 g) and N-ethyl-2-hydroxy-N,N-dimethylethanaminium bromide (0.16 g) were added into water (10 mL) under ultrasonic homogenizer until a uniform suspension was formed. Secondly, 0.52 g of copper (II) chloride dihydrate was dissolved into water (10 mL). Then two solutions were mixed with constant stirring at room temperature for 30 min. The obtained HKUST-1 particles (0.075 g) were collected by centrifugation and then, washed with ethanol/water and dried. 

#### 2.2.4. Synthesis of Imazalil IL-3@HKUST-1

Then, 0.60 g of imazalil was placed in a three-neck bottle containing dehydrated HKUST-1 (0.16 g) and ethanol (30 mL). The mixture was stirred at room temperature for about 10 h. Approximately 0.50 g of 1-bromododecane was added to the mixture and stirred for additional 28 h under reflux. The imazalil IL-3@HKUST-1 solid product was obtained by centrifugation, washed with water/ethanol and vacuum-dried at 50 °C for 24 h. 

### 2.3. Characterization

SEM (Hitachi S4800, Tokyo, Japan) was used to characterize the morphology and structures of HKUST-1. Thermogravimetric analysis (TGA) of the nanocapsules was performed to determine their thermal stability using an SDT Q600 (TA Instruments-Waters LLC, New Castle, DE, USA) analyzer in the temperature range of 50 °C to 800 °C at a heating rate of 10 °C min^−1^. The N_2_ sorption measurement was maintained at 77 K after degassing at 120 °C for 6 h by an ASAP 2010 surface area analyzer (Micromeritics Instrument Corporation, Norcross, GA, USA). The chemical compositions of the specimens were measured by XPS (Perkin Elmer, Waltham, MA, USA).

### 2.4. Antibacterial Activity

The antibacterial activity of imazalil, imazalil IL-3, HKUST-1 and imazalil IL-3@HKUST-1 against *Pseudomonas syringae* pv. *lachrymans* and *Xanthomonas campestris* pv. *campestris* was investigated by a broth dilution method. The *Pseudomonas syringae* pv. *lachrymans* and *Xanthomonas campestris* pv. *campestris* were rejuvenated at 28 °C in the Luria–Bertani (LB) broth and shaken at 180 rpm for 12 h; and then, the obtained bacterial suspension was diluted to an optical density of about 0.4 at 600 nm (OD 600) for inoculation. Next, different concentrations of imazalil, imazalil IL-3, HKUST-1 and imazalil IL-3@HKUST-1 were incubated with the above bacterial suspensions, and the OD value at 600 nm was monitored in real time to test the antibacterial properties. According to the probit analysis, the EC_50_ values for the bacterial bioassay were determined using the log dose–response curves.

### 2.5. Antifungal Activity

The antifungal activity of HKUST-1, imazalil IL-3 and imazalil IL-3@HKUST-1 against *Alternaria brassicae* and *Botrytis cinerea* was investigated by the growth rate method. Briefly, different concentrations of HKUST-1, imazalil IL-3 and imazalil IL-3@HKUST-1 suspensions prepared in a potato dextrose agar medium (1:100 ratio) were mixed and poured into a sterile plate. The application dosages of HKUST-1, imazalil IL-3 and imazalil IL-3@HKUST-1 to the two fungi were 25, 50, 100, 200 and 400 mg L^−1^. Meanwhile, the blank plates were used as the control. After rejuvenation, the bacterial cake (diameter 5 mm) was inoculated in the above mixed plate and cultured at 28 °C for 3~4 days. Finally, the percentage of control efficiency was calculated by the diameter formed by colonies on the culture plate. According to the probit analysis, the EC_50_ values for the fungi bioassay were determined using the log dose–response curves.

### 2.6. Safety to Plants

The safety of HKUST-1, imazalil IL-3 and imazalil IL-3@HKUST-1 was tested on Chinese cabbage (*B. rapa* L.var.*pekinensis*) in the greenhouse of China Agricultural University. The average day/night temperature was 25/18 °C, the photoperiod was 15/9 h and the humidity was at 60–80% during the whole experiment. Cabbage seeds were sown in flower pots with a diameter of 8 cm. At the three-leaf stage of cabbage, different treatments were sprayed with 15.0 mL of suspension using a microaerosol sprayer. The application dosages of HKUST-1, imazalil IL-3 and imazalil IL-3@HKUST-1 were 300, 600 and 900 mg L^−1^, respectively, and the treatment with water was used as a control. After 7 days, the leaf SPAD value and fresh weight of each cabbage seedling were measured, and the safety was evaluated by comparing with the control fresh weight and the leaf SPAD value of cabbage seedlings.

### 2.7. Drug Release from Imazalil IL-3@HKUST-1

The imazalil-release behavior was studied under various pH conditions (5, 7, 9) (25 °C) over 120 h. At each testing point, the concentration of the imazalil (1 mL) solution was determined by using a previously reported spectroscopic method; imazalil concentration was determined using high performance liquid chromatography (LC20AT Shimadzu, Tokyo, Japan). Imazalil release data were also assessed with the zero-order, first-order, Higuchi and Ritger-Peppas models by Equations (1)–(4), respectively:*M_t_*/*M*_0_ = *kt* + *b*(1)
*M_t_*/*M*_0_ = 1 − *e*^−*kt*^(2)
*M_t_*/*M*_0_ = *kt*^0.5^ + *b*(3)
*M_t_*/*M*_0_ = *kt^n^*(4)
where *M_t_*/*M*_0_ is the ratio of imazalil released from imazalil IL-3@HKUST-1 at time *t*, *k* is the rate constant and *n* is the diffusion parameter.

### 2.8. Statistical Analysis

Data were expressed as mean ± standard deviation. SPSS (SPSS, Inc., Chicago, IL, USA) was used for statistical analysis. *p* < 0.05 was considered statistically significant.

## 3. Results and Discussion

### 3.1. Fabrication and Characterization of Imazalil IL-3@HKUST-1

Firstly, HKUST-1 was prepared rapidly by benzenetricarboxylic acid and Cu(II) salt in an aqueous solution at room temperature. As shown in Figure 1A, the HKUST-1 particles are octahedrons, with an average diameter of approximately 1200 nm and well dispersed. Figure 1B shows the representative N_2_ adsorption/desorption isotherms for the synthesized HKUST-1 and imazalil IL-3@HKUST-1, respectively. A hysteresis is observed from the desorption curve of HKUST-1, which demonstrates the existence of microporous. The average pore size of HKUST-1 is 0.8–0.9 nm, and the total pore volume of imazalil IL-3@HKUST-1 (0.40 cm^3^ g^−1^) is obviously lower than that of HKUST-1 (0.93 cm^3^ g^−1^). The pore volume of HKUST-1 (0.56 cm^3^ g^−1^) is also higher than that of imazalil IL-3@HKUST-1 (0.16 cm^3^ g^−1^). These results demonstrated that imazalil IL was successfully loaded in the HKUST-1.

The surface characteristics of the synthesized samples were analyzed by X-ray photoelectron spectra (Figure 1C). For HKUST-1, the binding energies of Cu 2p_1/2_ and Cu 2p_3/2_ were 954.1 and 934.1 eV, respectively (Figure 1D). For imazalil IL-3@HKUST-1, the binding energies of Cu 2p_1/2_ and Cu 2p_3/2_ were a little bigger than that of the pristine HKUST-1, which demonstrated that the coordination environment of the Cu^2+^ center was changed because of the Cu–N (imazalil IL) interaction. The powder X-ray diffraction pattern (Figure 1E) showed pesticide incorporation did not affect the crystal structure of HKUST-1. Similar results were reported by Alavijeh and Liang et al. [30,31]. To assess the loading efficiency of imazalil IL-3@HKUST-1, thermogravimetric analysis was used (Figure 1F). The weight loss below 165 °C was mainly attributed to the evaporation of physically attached water in the particles, and the imazalil IL-3@HKUST-1 experienced two-step main decomposition profiles. The first weight loss step in the range of 200–320 °C could be due to the decomposition of imazalil IL-3 inside the nanocavities of HKUST-1. The second decomposition step at >320 °C was ascribed to the complete collapse of MOF. The total weight losses of HKUST-1 and imazalil IL-3@HKUST-1 from 35 to 800 °C were 24.25% and 20.18%, respectively. Therefore, the loading efficiency of imazalil IL-3 and copper in the HKUST-1 was, respectively, 4% and 5.49% from Table 1.

### 3.2. In Vitro Bactericidal Activity Test

The half maximal effective concentration (EC_50_) testing was used to estimate the antimicrobial properties of imazalil ILs, HKUST-1 and imazalil IL-3@HKUST-1 against *Pseudomonas syringae* pv. *lachrymans* and *Xanthomonas campestris* pv. *campestris* (the cause of black rot). For synthesized pesticides and materials, the density of the bacteria showed a decrease in a typical dose-dependent manner (Appendix A). According to the results (Table 2), the imazalil IL exhibited a strong inhibitory effect against the two pathogenic bacteria. The copper content of the imazalil IL-3@HKUST-1 is 5.49%. The water and low pH value could lead to slow hydrolysis of the metal-linker bond in HKUST-1, followed by a partial collapse of the framework, resulting in releasing copper ions and active substances gradually. For *Pseudomonas syringae* pv. *lachrymans* and *Xanthomonas campestris*, the long alkyl chain of ILs has the facility to destroy the cellular membrane. After loading imazalil IL-3, imazalil IL-3@HKUST-1 exhibited more promising antibacterial activities than pure HKUST-1 and individual imazalil IL-3. The co-toxicity coefficients were 102.75 and 105.26 for the EC_50_, which indicates good antibacterial activity against the two pathogenic bacteria.

### 3.3. In Vitro Fungicidal Activity Test

For the two pathogenic fungi, the test compounds displayed a stronger inhibitory effect against *Botrytis cinerea* than *Alternaria brassicae* (Figure 2). At the studied concentration, HKUST-1 did not have a significant impact on two pathogenic fungi, whereas imazalil IL-3 leaded to a great inhibitory effect. Meanwhile, after loading imazalil IL-3, imazalil IL-3@HKUST-1 had an inhibitory effect on the two pathogenic fungi. Moreover, the co-toxicity coefficients were 174.89 and 162.46 for the EC_50_, which indicates synergism at the exposure periods for the two pathogenic fungi (Table 3). Benefiting from the slow and constant release of copper ions and imazalil IL-3, imazalil IL-3@HKUST-1 exhibited a better effective residual microbicidal activity than imazalil (Appendix A).

### 3.4. In Vitro Drug Release Testing

The obtained encapsulation system can respond to external stimulus. Figure 3 shows the release behaviors of the imazalil IL-3@HKUST-1 nanocapsules. There was more than 80% imazalil released from the imazalil IL-3@HKUST-1 nanocapsules after 72 h in PBS solution (pH 5); however, only 30.01% imazalil released from imazalil IL-3@HKUST-1 nanocapsules after 120 h in PBS solution (pH 9). At the same time, about 82.79% imazalil was finally released from the imazalil IL-3@HKUST-1 nanocapsules in PBS solution (45 °C); however, only 57.81% imazalil was released from the imazalil IL-3@HKUST-1 nanocapsules after 120 h in PBS solution (45 °C). These showed the cumulative imazalil release rate from the nanocapsules in the release media of pH 5 at room temperature. The cumulative release rates of imazalil were in the order pH 5 > pH 7 > pH 9, which demonstrated that the hydrolysis rate of the MOFs metal construction was low in medium and alkalinity condition.

### 3.5. Mathematical Model

The cumulative release data of imazalil were fitted to the Riger-Peppas model (Table 4), and there was a good correlation of the release curve of imazalil with the kinetic equation *Mt/M*_0_ = *kt^n^* (r > 0.95), where n values revealed the mechanism of the release. In the kinetic equation of the imazalil IL-3@HKUST-1 nanocapsules under the condition of pH 9, the n value was 0.36, which indicated that the release of imazalil was a Fickian transport at pH 9. The n values were 0.54 and 0.57 at pH 5 and 7, respectively; the release of imazalil occurred as a non-Fickian (anomalous) release, which indicated that the release of imazalil was driven by the swelling or degradation of shell and diffusion. The T_50_ (34.42 h) was the shortest at pH 5, which manifested that the release rate of imazalil from the imazalil IL-3@HKUST-1 nanocapsules was the fastest; when the value of T_50_ (510.25 h) under the condition of pH 9 was much higher than that (116.02 h) under the condition of pH 7.0. Therefore, the release of an active ingredient from the imazalil IL-3@HKUST-1 nanocapsules was triggered by the acidic condition.

### 3.6. Safety Evaluation of Imazalil IL-3@HKUST-1

The safety of pesticides based on MOFs as a carrier cannot be ignored in crops. In addition, the cooper(II) released by MOFs may also be harmful to crops. The safety evaluation to Chinese cabbage (*Brassica rapa pekinensis*) of chlorophyll content, imazalil IL and HKUST-1 in the concentrations of 300, 600 and 900 mg/L was conducted in the greenhouse. The results showed that the synthesized imazalil IL-3@HKUST-1 had no significant effect on the fresh weight and SPAD value of the tested plant leaf. Moreover, imazalil IL and HKUST-1 are also safe to control the bacterial pathogens in cabbages, and possibly other copper-tolerant crops (Table 5). MOFs degrade slowly in vivo and can slowly release copper ions and pesticides; thus, causing no harm to crops when controlling plant diseases.

## 4. Conclusions

In conclusion, we firstly conceived a rapid room-temperature synthesis strategy that led to porous HKUST-1, with cholinium salt as the deprotonation agent and template. A novel antimicrobial system based on imazalil IL-encapsulated hierarchical porous MOF (imazalil IL-3@HKUST-1) was established via the ship-in-bottle technique. The prepared imazalil IL-3@HKUST-1 had excellent antibacterial properties toward *Pseudomonas syringae* pv. *lachrymans* and *Xanthomonas campestris* pv. *campestris*. Meanwhile, it exhibited strong antifungal activities toward *Alternaria brassicae* and *Botrytis cinerea*. Imazalil IL-3@HKUST-1 could achieve a synergistic effect in effectively inhibiting pathogenic fungi and bacteria. Thus, the imazalil IL-3@HKUST-1 is a promising antimicrobial for integrated plant diseases management.

## Data Availability

Not applicable.

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
