# Peer review of "Encapsulation of Imazalil in HKUST-1 with Versatile Antimicrobial Activity"

_nanomaterials, 2022, doi:10.3390/nano12213879_

Round 1
Reviewer 1 Report (New Reviewer)
Dong et al have shown a new method for encapsulating a pesticide into a porous system. The beauty of this study is using the slow degradation on HKUST in an aqueous environment to slowly release the pesticide.
The study is very good and consistent. Very interesting in my opinion is the MOF synthesis. Could you calculate the yield please.
I only have some short questions/remarks concerning the adsorption measurements.
The authors state a hysteresis for HKUST-1 bulk material due to the mesopores of the MOF. However, a hysteris should occur above 0.42 p/p0 and not below. Furhtermore, the desorption branch cannot be distinguished from the adsorption branch. Please use empty and filled symbols for desorption and adsorption respectively. Please state which model was used to calculate the pore size distribution. Is there any benefit to calculate this? Why was the Langmuir surface are be calculated? This model is not applicable for mesoporous systems. Please use the BET model or do not state the surface area at all. The pore volume at 0.99 p/p0 is way more interesting since this value gives you a glimpse about the loading/filling of your pores.
Please add a reference of HKUST-1 in your PXRD graph.
What represents the dash line in figure 1F?
Please increase the export quality of all your figures!
Author Response
Dong et al have shown a new method for encapsulating a pesticide into a porous system. The beauty of this study is using the slow degradation on HKUST in an aqueous environment to slowly release the pesticide.
The study is very good and consistent. Very interesting in my opinion is the MOF synthesis. Could you calculate the yield please.
I only have some short questions/remarks concerning the adsorption measurements.
- The authors state a hysteresis for HKUST-1 bulk material due to the mesopores of the MOF. However, a hysteris should occur above 0.42 p/p0 and not below. Furhtermore, the desorption branch cannot be distinguished from the adsorption branch. Please use empty and filled symbols for desorption and adsorption respectively. Please state which model was used to calculate the pore size distribution. Is there any benefit to calculate this? Why was the Langmuir surface are be calculated? This model is not applicable for mesoporous systems. Please use the BET model or do not state the surface area at all. The pore volume at 0.99 p/p0 is way more interesting since this value gives you a glimpse about the loading/filling of your pores.
Response: The pore size of the samples was calculated by using density functional theory (DFT), respectively. Pore-size distribution of HKUST-1 showed that the average pore size of HKUST-1 was 0.8-0.9 nm, which is attributed to the microporous structure of HKUST-1. In the paper, mesoporous has been revised to microporous. The Langmuir surface of the particles was deleted in the revised manuscript.
- Please add a reference of HKUST-1 in your PXRD graph.
Response: In this manuscript, the patterns of imazalil IL-3@HKUST-1 was matched to the PXRD pattern of HKUST-1, indicating that pesticide incorporation would not change the crystallinity of the MOFs. Similar results were reported by Alavijeh and Liang et al. [30. Y. Liang, S. Wang, H. Jia, Y. Yao, J. Song, H. Dong, Y. Cao, F. Zhu, Z. Huo, Pectin functionalized metal-organic frameworks as dual-stimuli-responsive carriers to improve the pesticide targeting and reduce environmental risks, Colloids Surf., B, 219 (2022) 112796. 31. R. Karimi Alavijeh, K. Akhbari, Biocompatible MIL-101 (Fe) as a smart carrier with high loading potential and sustained release of curcumin, Inorg. Chem., 59 (2020) 3570-3578.]
- What represents the dash line in figure 1F?
Response: The dash lines were deleted in Figure 1F.
- Please increase the export quality of all your figures!
Response: The export quality of the figures was increased in the revised manuscript.

Reviewer 2 Report (New Reviewer)
The authors have prepared imazalil encapsulated porous HKUST-1 as a novel antimicrobial agent. The Imazalil IL-3@HKUST-1 has exhibited synergistic effect for the inhibition of pathogenic fungi and bacteria. The authors have also demonstrated that IL@HKUST-1 has revealed slow and constant release of imazalil for better sustainability and microbicidal activity than imazalil. Overall, this work can inspire more material design ideas of MOFs for antibacterial application. Therefore, I would like to recommend this work to publish in Nanomaterials. Below are some comments for the authors.
1. The diameter of HKUST-1 particles should be calculated and the histogram of the diameter distribution should be provided in the supporting information.
2. For powder X-ray diffraction pattern of Figure 1(E), the characteristic peaks of HKUST-1 and imazalil IL-3@HKUST-1 should be labeled in the spectra.
3. This paper would be more impressive if the authors could provide the SEM images of bacteria after incubation with imazalil IL-3@HKUST-1 to see the damage of bacteria.
4. For the introduction “Therefore, it is urgent to develop a new strategy to overcome the antimicrobial resistance...”, more references could be cited to broaden the introduction.
https://doi.org/10.3390/nano10061123
Author Response
The authors have prepared imazalil encapsulated porous HKUST-1 as a novel antimicrobial agent. The Imazalil IL-3@HKUST-1 has exhibited synergistic effect for the inhibition of pathogenic fungi and bacteria. The authors have also demonstrated that IL@HKUST-1 has revealed slow and constant release of imazalil for better sustainability and microbicidal activity than imazalil. Overall, this work can inspire more material design ideas of MOFs for antibacterial application. Therefore, I would like to recommend this work to publish in Nanomaterials. Below are some comments for the authors.
- The diameter of HKUST-1 particles should be calculated and the histogram of the diameter distribution should be provided in the supporting information.
Response: Thanks for your suggestions. In our previous research, the average size and PDI of HKUST-1 particles were reported, which were 1690 nm and 0.213, respectively [C. Fan, H. Dong, Y. Liang, J. Yang, G. Tang, W. Zhang, Y. Cao, Sustainable synthesis of HKUST-1 and its composite by biocompatible ionic liquid for enhancing visible-light photocatalytic performance, Journal of Cleaner Production, 208 (2019) 353-362.]. In this paper, relative data has not been added.
- For powder X-ray diffraction pattern of Figure 1(E), the characteristic peaks of HKUST-1 and imazalil IL-3@HKUST-1 should be labeled in the spectra.
Response: The characteristic peaks of HKUST-1 and imazalil IL-3@HKUST-1 were labeled in the revised manuscript.
- This paper would be more impressive if the authors could provide the SEM images of bacteria after incubation with imazalil IL-3@HKUST-1 to see the damage of bacteria.
Response: Thanks for your suggestions. In this experiment, SEM images of bacteria were not determined, in our following works, we will try to make our study more impressive by the SEM images of bacteria.
- For the introduction “Therefore, it is urgent to develop a new strategy to overcome the antimicrobial resistance...”, more references could be cited to broaden the introduction.
Response: The reference was added in the revised manuscript(S. Yougbaré, C. Mutalik, D.I. Krisnawati, H. Kristanto, A. Jazidie, M. Nuh, T.-M. Cheng, T.-R. Kuo, Nanomaterials for the photothermal killing of bacteria, Nanomaterials, 10 (2020) 1123. https://doi.org/10.3390/nano10061123).

This manuscript is a resubmission of an earlier submission. The following is a list of the peer review reports and author responses from that submission.
Round 1
Author Response
Response to Reviewers’ Comments:
1. Lines78, 98 and 100: Sentences cannot begin with “and”. Response: Thank you for your comments. The sentences were corrected in the revised manuscript.2. Line 100: Should be “washed” (not washing). Response: The mistake has been corrected.3. Line 102: Equimolar with respect to what? Response: The same molar amount of 1-bromododecane as imazalil IL-3. That has been revised.4. Line 104: omit the “an”. Response: That has been revised.5. Line 122: with what? Response: “from 50 ℃ to 800 ℃ with a heat rate of 10 ℃ min−1.” has been revised to “in the temperature range of 50 ℃ to 800 ℃ with a heating rate of 10 ℃ min−1.”6. Line 143: Omit the two “and”. Response: That has been revised.7. Line 153: Should be: “The imazalil-release behavior was studied under various conditions of pH and temperatures….” Response: That has been revised.8. Line 159: Should be “first”. Response: That has been revised.9. Eq. 1-4: Mz should be M0? Response: That has been revised.10. Line 164: Should be “ratio” (not percentage). For expressing this ratio as percentage, it should be multiplied by 100%. Response: That has been revised.11. Line 171: (a) Why was is synthesized rapidly? Perhaps it was ment “reacted rapidly?” (b) It was synthesized from Cholinium salt and what else? (c) Cholinium salt was not mentioned in the experimental section. Response: As shown in Scheme 1, we developed a simple and fast synthesis strategy that resulted in hierarchical porous HKUST-1 with cholinium salt as the deprotonation agent and template. The information about cholinium salt was supplied in the supplementary materials. 12. Line 174: How can the pore diameter be 1200 nm (1.2 µm) when an entire single particle is only ca. 1 µm? Response: The average particle size of the synthetic HKUST-1 is indeed 1200 nm.13. Line 177: The hysteresis in Fig. 1B is very slight. Response: Similar results have been reported by 14. Line 194: Should be “synthesized from”. Response: That has been revised.15. Lines 207-209: (a) How were the loading efficiencies calculated from the data in Table 1? (b) Should be spelled “copper”. (c) The loading values should be rounded to 4 and 1. Response:(a) Firstly,compare the difference of chloride content between HKUST-1 and the composite IL-@HKUST-1. Then, the imazalil IL-3 percentages in composite IL- @HKUST-1was calculated Indirectly based on the fixed value of chlorine in structure of the imazalil IL-3.(b) and (c)That has been revised.16. Section 3.2: It is said that the imazalil exhibited stronger inhibitory effect than HKUST-1 against the two types of bacteria, based in the higher EC50 values in Table 2. This value is also higher than that for the composite IL-@HKUST-1, so why was it concluded that the composite exhibits more promising antibacterial activity? This section requires more clarity and rephrasing. Response: The higher EC50 of the composite IL- @HKUST-1 is due to the low pesticide loading efficiency of Izolil IL-3 and copper in HKUST-1 at 4.07% and 5.49%, respectively. The ecotoxicity coefficient of the composite IL-@HKUST-1 is greater than 100, so composite IL- @HKUST-1 is considered to have more promising antibacterial activity. That has been revised.17. Section 3.3: Here, high EC50 values obtain with HKUST-1 were interpreted as insignificant effect, whereas low EC50 for IL-3 was considered as great effect. In both Tables 2 and 3 the EC50 values for the complex are intermediate, so where is the synergistic effect? This should be better explained.Response: That has been revised.18. Tables 2 &3: (a) What is the meaning of a ratios in these tables? Ratios between what? (b) The numbers should not be given with four significant figures. Response: The meaning of a ratios in these tables was added in in the revised manuscript. The numbers in Table 2 and 3 has been revised. 19. Line 342: A legend is required for Scheme 1 to explain the various shapes.Response: The legends have been added in Scheme 1.

Reviewer 2 Report
Manuscript well written, and topic is of broad interest. Authors are demonstrating utility of the combined products to improve micobicidal activity, thus in conclusions care should be taken to not 'overstate' the results. See suggested sentence changes in manuscript comments. CU release in plants can also be a health concern, especially in cabbages and other vegetables, thus references should be added pg.6, Line 268, reference: Nitika Sharma et al 2021 IOP Conf. Ser.: Earth Environ. Sci. 889 012071, and ref: Yang XE, Long XX, et al. Assessing copper thresholds for phytotoxicity and potential dietary toxicity in selected vegetable crops. J Environ Sci Health B. 2002 Nov;37(6):625-35. doi: 10.1081/PFC-120015443. PMID: 12403270. Modify wording Line 273, -"... to control bacterial pathogens in cabbages, and possibly other copper-tolerant crops." Modify wording in Conclusions - Line 284 delete 'can be used as', and use 'is' a promising antimicrobial.........
Author Response
1. Modify wording Line 273, -"... to control bacterial pathogens in cabbages, and possibly other copper-tolerant crops." Response: The sentence has been corrected.2. Modify wording in Conclusions - Line 284 delete 'can be used as', and use 'is' a promising antimicrobial Response: That has been revised.
Reviewer 3 Report
In the research article entitled “Encapsulation of imazalil ionic liquid in HKUST-1 with versatile antimicrobial activity”, the authors have systematically investigated liquid-encapsulation of antimicrobial imazalil in the metal-organic framework (IL-3@HKUST-1) and its antimicrobial activity. Authors have demonstrated control and sustained release of copper and imazalil IL-3 from imazalil IL-3@HKUST-1. The manuscript lacks a few important topics which need to be addressed such as stability of the formulation, drug loading capacity, etc. The article has many grammatical and sentence errors, and the language organization needs to be improved. For these reasons, I conclude that the paper should undergo minor revisions.
1. In the Abstract section, the authors can highlight keys and more emphasis on the finding (numerical values) and its implication may be mentioned in the abstract like the IC50 value of all tested microbes, and kinetics study details.
2. Good introduction, Still, it should be improved. To make the introduction more substantial, the author should provide several updated recent references to substantiate the claim made.
3. The stability of the formulation needs to be evaluated using storage which is important. authors may refer to
https://doi.org/10.1016/j.molliq.2021.116623
https://doi.org/10.1016/j.ejps.2017.12.028
4. Drug loading capacity of IL-3@HKUST-1 needs to determine.
5. There are many grammatical and sentence errors in the article, and the language organization needs to be improved.
6. Please improve the conclusion with clear quantitative findings and more emphasis on finding and its implication may be mentioned in the conclusion section.
Author Response
- In the Abstract section, the authors can highlight keys and more emphasis on the finding (numerical values) and its implication may be mentioned in the abstract like the IC50 value of all tested microbes, and kinetics study details.
Response: Thank you for your comments. That has been revised.
- Good introduction, Still, it should be improved. To make the introduction more substantial, the author should provide several updated recent references to substantiate the claim made.
Response: That has been revised.
- The stability of the formulation needs to be evaluated using storage which is important. authors may refer tohttps://doi.org/10.1016/j.molliq.2021.116623, https://doi.org/10.1016/j.ejps.2017.12.028
Response: The release of the active ingredient from imazalil IL-3@HKUST-1 nanoparticles can be triggered under acidic conditions.
- Drug loading capacity of IL-3@HKUST-1 needs to determine.
Response: The pesticide loading efficiency of imazalil IL-3 and copper in the HKUST-1 was respectively 4.07% and 5.49% from Table 1.
- There are many grammatical and sentence errors in the article, and the language organization needs to be improved.
Response: That has been revised.
- Please improve the conclusion with clear quantitative findings and more emphasis on finding and its implication may be mentioned in the conclusion section.
Response: The conclusion has been revised.

Reviewer 4 Report
The manuscript entitled "Encapsulation of imazalil ionic liquid in HKUST-1 with versatile antimicrobial activity" by T. Tang et al. is a nice contribution in the field of MOFs with biological activity. Overall, the paper is well done, but some information needs to be improved:
-Introduction section- delete the first paragraph, lines 31-46, possibly to be moved where the antifungal activity is discussed.
-lines 51-56 in Introduction are somehow confusing, it would be better to rephrase this content.
-the motivation to choose the HKUST 1 MOF for this application is not really clear presented-lines 60-63.
-lines 78-80 the ship -in -bottle technique is not clear presented in Scheme 1. Some additional information is need to the proposed scheme.
-lines 94-104-in some cases authors mentioned the number of mols in other case the amount, please insert both data. Please mention the total amount of the MOFs particles that was obtained.
FTIR technique must be applied to also confirm the total deprotonation of the carboxylic groups of the ligand and the successful complexation of copper ions. The coordination mode of copper can be also be done by this technique. Otherwise is difficult to discuss the encapsulation of drugs in MOF, especialy that the authors claim some new interactions between copper and the encapsulated drug. PXRD supported these interactions? some comments must be added at this part.
Why the drug release was studied at different pH values?especially since the release is of particular interest to plants. Did authors calculate the encapsulation efficiency?
-line 178-the authors claim a surface of MOF of 2045 m2/g. Did authors perform the activation of MOF? because they also mentioned a thermal instability beginning with 35 oC when the drug is present in the network. The thermal data befor encapsulation must be added and discussed.
-at the release studies, dis authors also observed the release of copper aions? what concentration? it is beneficial for the viability of plants? please add some comments regarding this.
Based on these comments I suggest the publication of this paper after minor revision.
Author Response
1. Introduction section- delete the first paragraph, lines 31-46, possibly to be moved where the antifungal activity is discussed. Response: The sentences were rephrased.2. lines 51-56 in Introduction are somehow confusing, it would be better to rephrase this content. Response: The sentences were rephrased.3. the motivation to choose the HKUST 1 MOF for this application is not really clear presented-lines 60-63.Response: The sentences were rephrased.4. lines 78-80 the ship -in -bottle technique is not clear presented in Scheme 1. Some additional information is need to the proposed scheme.Response: The sentences were rephrased.5. lines 94-104-in some cases authors mentioned the number of mols in other case the amount, please insert both data. Please mention the total amount of the MOFs particles that was obtained.Response: The data were added in the revised manuscript..6. FTIR technique must be applied to also confirm the total deprotonation of the carboxylic groups of the ligand and the successful complexation of copper ions. The coordination mode of copper can be also be done by this technique. Otherwise is difficult to discuss the encapsulation of drugs in MOF, especialy that the authors claim some new interactions between copper and the encapsulated drug. PXRD supported these interactions? some comments must be added at this part.Response: Here, XRD was measured to prove the MOF, and TG was measured to claim the drug loading. 7. Why the drug release was studied at different pH values?especially since the release is of particular interest to plants. Did authors calculate the encapsulation efficiency?Response: The drug release at different pH values was investigated, considering the more acidic microenvironment that appeared after the invasion of plant pathogenic bacteria. The pesticide loading efficiency of imazalil IL-3 and copper in the HKUST-1 was respectively 4.07% and 5.49% from Table 1. 8. line 178-the authors claim a surface of MOF of 2045 m2/g. Did authors perform the activation of MOF? because they also mentioned a thermal instability beginning with 35 oC when the drug is present in the network. The thermal data befor encapsulation must be added and discussed. Response: The HKUST-1 was finally activated under vacuum at 120 ℃for 16 h and stored in closed vials in an oven at 60 ℃ for further experiments. According to the results of TGA analysis, the HKUST-1 has good thermal stability under 180℃, and the weight loss of imazalil IL-3@HKUST-1 was mainly attributed to the weight loss of imazalil IL-3.9. at the release studies, dis authors also observed the release of copper aions? what concentration? it is beneficial for the viability of plants? please add some comments regarding this.Response: That has been revised.

Round 2
Reviewer 1 Report
The revised article includes many corrections that were introduced according to the comments and suggestions. However, it still suffers from stylistic and grammatical problems. Moreover, there are some lacks of clarity in the synthetic procedure and the relative efficiency of the antibacterial and antifungal agents. Using professional stylistic assistance could be beneficial to improve this manuscript. Here are some points to be corrected:
1. Abstract: a) Lines 3-4 (and also online 1 of section 3.1) “Was rapidly synthesized” means that the authors hurried up to perform the synthesis. I commented on it in the previous time that they probably meant “reacted rapidly”. b) It is not clear what was released: the entire imazalil IL@HKUST-1 complex or just imazalil IL from the imazalil IL@HKUST-1 complex? c) Line 9: Should be “sustainable” or “better sustainability and microbicidal activity”.
2. P. 2, line 2: A comma is needed after (MOFs).
3. P. 3, line 3: What is the meaning of “development prospects”? Perhaps should be just “potential”?
4. P. 3, line 5: What is the meaning of ” 2.0 mmol of 1-bromododecane as imazalil IL-3 was added:? Does it mean similar amounts?
5. P.3, line 8: What else is given in the ESI? (”also”)
6. P. 3, line 13: “Has been widely (what?) to control…”
7. P. 3, line 14: “…and pear scab”
8. P. 3, line 18: Should be “A realm”. It is not “another” because no such example was given before.
9. P. 3, line 14: Should be “at heating rate”
10. Section 3.1, line 7: The hysteresis in Fig. 1B is very slight and does not seem to be meaningful.
11. P.5, lines 18-19: The loading values should be rounded to 4 and 1, respectively. This has not been corrected.
12. Section 3.2, line 3: Should be” “and Xanthomonas…”
13. Section 3.2, line 4: Should be “For the synthesized…”
14: Section 3.2: line 6: Should be ”…against the two…”
15: Section 3.2: line 11: The meaning of “disturb” is unclear. It should be” destroy” or” break”.
16. Section 3.2: line 16: Should be “…the two pathogenic…”
17: Section 3.3, last two lines: a) “After 6 days after treatment??” b) Should be “…system exhibited effective fungal inhibition.”
18. Scheme 1: No legend or explanations was added, only the names of the reagents. What are the meanings of the blue square with the red circle, the green lines and the arrows? Where is CuCl2 involved in the synthesis?

Author Response
Response to reviewers
Dear editors,I am the author of the manuscript “Encapsulation of imazalil ionic liquid in HKUST-1 with versatile antimicrobial activity” written by Hongqiang Dong, Yuke He, Chen Fan, Zhongqiang Zhu, Chunrong Zhang, Xinju Liu, Kun Qian, Tao Tang. Thank you for your consideration and help for giving us so much valuable comments and reviewers advice, they will do very good to my paper and research work in the futrue. Based on these comments and suggestions, we have made modification carefully on the original manuscript. We are looking forward to having chance to publish our work on your journal. Thank you for help again!Best regards!Kun Qian 1. According to comments of editor and reviewers1. Abstract: a) Lines 3-4 (and also online 1 of section 3.1)“Was rapidly synthesized”means that the authors hurried up to perform the synthesis. I commented on it in the previous time that they probably meant “reacted rapidly”. b) It is not clear what was released: the entire imazalil IL@HKUST-1 complex or just imazalil IL from the imazalil IL@HKUST-1 complex? c) Line 9: Should be “sustainable” or “better sustainability and microbicidal activity”.Answer: a) “Was rapidly synthesized” means that this reacts rapidly. That has been revised. b) There is imazalil IL from the imazalil IL@HKUST-1 complex. c)That has been revised.2. P. 2, line 2: A comma is needed after (MOFs).Answer: That has been revised.3. P.3, line 3: What is the meaning of “development prospects”? Perhaps should be just “potential”?Answer: That has been revised.4. P. 3, line 5: What is the meaning of” 2.0 mmol of 1-bromododecane as imazalil IL-3 was added:? Does it mean similar amounts?Answer: It mean similar amounts.5. P.3, line 8: What else is given in the ESI? (”also”)Answer: That has been revised.6. P. 3, line 13: “Has been widely (what?) to control…”Answer: Imazalil has been widely uesd to control...7. P. 3, line 14: “…and pear scab”Answer: That has been revised.8. P. 3, line 18: Should be “A realm”. It is not “another” because no such example was given before.Answer: That has been revised.9. P. 3, line 14: Should be “at heating rate”Answer: That has been revised.10. Section 3. 1, line 7: The hysteresis in Fig. 1B is very slight and does not seem to be meaningful.Answer: The hysteresis in Fig.1Bdemonstrates the existence of mesopores.11. P.5, lines 18- 19: The loading values should be rounded to 4 and 1, respectively. This has not been corrected.Answer: That has been revised.12. Section 3.2, line 3: Should be” “and Xanthomonas…Answer:That has been revised.13. Section 3.2, line 4: Should be “For the synthesized…” Answer:That has been revised.14. Section 3.2: line 6: Should be ”…against the two…”Answer: That has been revised.15: Section 3.2: line 11: The meaning of “disturb” is unclear. It should be” destroy” or” break” .Answer:That has been revised.16. Section 3.2: line 16: Should be “…the two pathogenic…”Answer:That has been revised.17: Section 3.3, last two lines: a) “After 6 days after treatment??” b) Should be “…system exhibited effective fungal inhibition.”Answer:a)There is “After 6 days treatment”.b)That has been revised.18. Scheme 1: No legend or explanations was added, only the names of the reagents. What are the meanings of the blue square with the red circle, the green lines and the arrows? Where is CuCl2 involved in the synthesis?Answer:That has been revised.

Round 3
Reviewer 1 Report
This manuscript has been revised and can now be published.
There are two typographic errors to be corrected:
1) Abstract, line 8: Should be :"from"
2) P. 2, line 14: Should be "used".